# Novel 2-Amino-1,4-Naphthoquinone Derivatives Induce A549 Cell Death through Autophagy

**DOI:** 10.3390/molecules28083289

**Published:** 2023-04-07

**Authors:** Hua-Yuan Tan, Feng-Ming Liang, Wen-Jing Zhang, Yi Zhang, Jun-Hao Cui, Yu-Yu Dai, Xue-Mei Qiu, Wen-Hang Wang, Yue Zhou, Dan-Ping Chen, Cheng-Peng Li

**Affiliations:** 1School of Pharmaceutical Sciences, Guizhou University, Guiyang 550025, China13436002029@163.com (F.-M.L.);; 2Guizhou Engineering Laboratory for Synthetic Drugs, Guizhou University, Guiyang 550025, China

**Keywords:** 2-amino-1,4-naphthoquinone, single crystal, autophagy

## Abstract

A series of 1,4-naphthoquinone derivatives containing were synthesized as anti-cancer agents and the crystal structure of compound **5a** was confirmed by X-ray diffraction. In addition, the inhibitory activities against four cancer cell lines (HepG2, A549, K562, and PC-3) were tested, respectively, and compound **5i** showed significant cytotoxicity on the A549 cell line with the IC_50_ of 6.15 μM. Surprisingly, in the following preliminary biological experiments, we found that compound **5i** induced autophagy by promoting the recycling of EGFR and signal transduction in the A549 cell, resulting in the activation of the EGFR signal pathway. The potential binding pattern between compound **5i** and EGFR tyrosine kinase (PDB ID: 1M17) was also identified by molecular docking. Our research paves the way for further studies and the development of novel and powerful anti-cancer drugs.

## 1. Introduction

Cancer is the second most common serious public health problem behind cardiovascular diseases [1]. The burden of the incidence rates and mortality of cancers is rapidly increasing worldwide [2]. For this reason, the discovery of new and effective anti-cancer drugs for malignant tumor treatment is an urgent task, and at the same time, the development of promising anti-cancer agents becomes one of the research hotspots that attracts the great attention of chemists and pharmaceutical scientists. Death patterns of tumor cell are regulated by multiple signaling pathways. Autophagy is a physiological cellular process that degrades and eliminates misfolded proteins and damaged organelles, which plays a vital role in adaptive starvation, development, cell death, and tumor inhibition [3,4]. The development of new drugs and the study of the mechanism that causes tumor cells’ death is a feasible approach for advances in cancer therapy.

The NCI (National Cancer Institute) demonstrated that quinone derivatives were a class of important active structures that are capable of inhibiting the proliferation of cancer cells. Among all the quinone structure compounds, naphthoquinone exhibits a more remarkable biological activity than the others, especially in terms of its anti-tumor activity. The 1,4-naphthoquinone structure, known as a kind of ubiquitous natural compounds [5,6,7,8], is a typical structure for the development of perspective drugs, which is usually separated and purified from natural plants (such as *xanthium sibiricum* [9]). Except for being used as the raw material for the synthesis of medicine and pesticides [10], 1,4-naphthoquinones also display a wide range of biological activities [11,12], such as anti-bacterial [13,14,15,16,17,18,19], anti-virus [20,21,22,23,24], anti-cancer [25,26,27], anti-malaria [28], treatment of protozoal disease [29], anti-Alzheimer’s disease [30], inhibition of atherosclerosis, elimination of free radicals [31], inhibition of platelet activity [32], trypanocidal [33], and anti-inflammatory [34]. So the structure of 1,4-naphthoquinone has amazing potential for different kinds of medical applications. Recently, various anti-cancer drugs containing 1,4-naphthoquinone structural units [35], such as doxorubicin, daunorubicin, atovaquone [36], buparvaquone [37], shikonin, etc. (Figure 1), were used for solid tumors therapy, including lung cancer, prostate cancer, and breast cancer, which achieved great therapeutic effects. In brief, 1,4-naphthoquinone derivatives are able to provide a new insight for the development of new anti-cancer drugs through the further modification of the potential leading compounds to enhance their biological activities [38].

Based on the remarkable biological activity of 1,4-naphthoquinone derivatives in inhibiting the growth of tumor cells, recently, an increasing number of pharmacists devoted themselves to the continuous innovation and design of promising 1,4-naphthoquinone structure drugs, the comprehensive study of the relationship between compound structures and biological activities, and the investigation of the mechanism concerning the inhibition of the growth of tumor cells. The previous research found that 1,4-naphthoquinone derivatives containing an amide structure enabled the improvement of anti-tumor activities significantly [39,40]. The amide group is regarded as a common active structural segment based on its anti-bacterial [41], insecticidal [42], anti-cancer [43] and other biological activities, which is often introduced to design drugs that are expecting to achieve great therapeutic effect. The introduction of an amide structure is able to increase the polarity and solubility of anti-tumor drugs. More importantly, the pharmacological effects of amino anti-tumor drugs are greatly improved based on the interaction between target proteins and molecules [44,45,46,47,48,49,50,51].

Inspired by the high activity of 1,4-naphthoquinone derivatives and the interaction ability of an amide structure, we designed and synthesized a series of two amino-substituted 1,4-naphthoquinone derivatives based on the principle of drug splicing, expecting to obtain promising anti-tumor agents. The crystal structure of the target compound **5a** was confirmed through X-ray diffraction, and the inhibitory activities of the synthesized compounds on A549 (non-small human lung cancer), PC-3 (prostate cancer l), K562 (chronic myeloid leukemia), and HepG2 (liver cancer) were studied by the MTT method, and compound **5i** was found to exhibit a high anti-proliferative activity against A549 cancer cells in vitro. Subsequently, **5i** was further investigated to analyze its anti-tumor mechanism of A549 cancer cells.

## 2. Results

### 2.1. Chemistry

In order to obtain the target compounds **5a**–**5u** with different substitutions and **9a**–**9g** containing the amide group, we designed a simple and feasible synthesis route, and the synthesis route in this work is shown in Figure 2. In the first synthesis route, intermediate **3** was prepared by the reaction of different substituted aniline with bromo-acetyl bromide, which was followed by a reaction with 2-hydroxy-1,4-naphthoquinone to synthesize the compounds **5a**–**5u** in a DMF solvent. The yield of the compounds **5a**–**5u** was 42.33–81.65% by the 1:1 ratio of cesium carbonate to 2-hydroxy-1,4-naphthoquinone. Secondly, compounds **9a**–**9g** that contained an amide structure were synthesized by 1,4-naphthoquinone and 3-amino-p-methylene acid, and copper acetate worked as the catalyst to obtain intermediate **8** [52]. Compounds **9a**–**9g** were prepared by reactions of intermediate **8** with different substituted aliphatic amines or anilines under alkaline conditions of the DIPEA (N,N-Diisopropylethylamine) and HATU (2-(7-Azabenzotriazol-1-yl)-N,N,N′,N′-tetramethyluronium hexafluorophosphate) mixture with yields of 43.24–60.64%. All the unreported synthesized compounds were confirmed by ^1^H NMR, ^13^C NMR, and HRMS spectrums, and the characterized materials are displayed in the Appendix A.

### 2.2. Crystal Structure Determination

In order to verify the feasibility of the synthetic route of the new compounds **5a**–**5u**, a single crystal of compound **5a** was cultured after the purification based on the gas phase diffusion method, and the crystal structure was obtained and identified by an X-ray single-crystal diffraction. The process was as follows: nearly 10–20 mg **5a** was dissolved in a DMSO, filtered through a 0.22 μm membrane, injected into a small culture bottle, sealed with a sealing membrane that left a gap of 4–5 pinholes, and finally, put into a large culture bottle with an appropriate amount of a low boiling solvent (water) at the bottom. What was noteworthy was that the sealed samples were placed at the indoor ambient temperature (about 25 °C) for fixation to enable the system to diffuse slowly. 

The single-crystal structure of compound **5a** was analyzed by Olex-2 software, and the results are shown in Figure 3. Compound **5a** was the colorless single crystal. The crystal data of C_17_H_13_NO_2_ (M = 263.290 g/mol) were as follows: orthorhoombic, space group P212121 (no. 19), a = 7.3985(3) Å, b = 12.1825(4) Å, c = 15.1213(5) Å, V = 1362.92(8) Å3, Z = 4, T = 273.15 K, μ(Cu Kα) = 0.680 mm*^−^*^1^, Dcalc = 1.278 g/cm^3^, 7003 reflections measured (9.32° ≤ 2Θ ≤ 33.14°), and 2353 unique (Rint = 0.0514, Rsigma = 0.0512), which were used in all the calculations. The final R1 was 0.0526 (I ≥ 2u(I)) and wR2 was 0.1819. 

### 2.3. Biological Activity

In this work, gefitinib (G) and 5-fluorouracil were used as the positive controls, and the in vitro cytotoxic activities of all the synthetic compounds against the A549, PC-3, K562, and HepG2 cell lines were evaluates in an MTT assay. The cytotoxic activities’ results of compounds **5a**–**5u** after 48 hours of treatment are shown in Appendix A), and most of the target compounds exhibited a good inhibitory activity against the A549, PC-3, HepG2, and K562 cancer cells. The anti-proliferative activity data of the compounds **9a**–**9g** are listed in Appendix A), and the results suggest that the propylamine-, butylamine-, and 3-fluorophenyl-substituted compounds displayed high bioactivities.

Based on the significant bioactivity of most of the target compounds against the A549 cells, we further selectively evaluated the anti-proliferative activity of these 1,4-naphthoquinone derivatives against cell carcinoma lines, and the results are summarized in Table 1. Most of the synthesized compounds showed IC_50_ (50% of the compounds that inhibit cancer cell growth) values below 20 µM. Moreover, the IC_50_ values of compounds **5a**, **5b**, **5i**, **5l**, **5m**, **5p,** and **5t** were below 10 µM, indicating the remarkable anti-tumor ability of these compounds. An interesting phenomenon was found in compounds **5a**-**5u** once they shared the same substituent groups R, and their activities were different at various substitution positions, such as compound **5d** (R = 2-Cl, 20.32 µM) > **5e** (R = 3-Cl, 23.45 µM) > **5f** (R = 4-Cl, 50 µM). According to the bioactivity data, we supposed that the design substituent at two or three sites of the target compounds enabled better anti-tumor activity than four of the site substituent. For the results of compounds **9a**–**9g**, an increase in the length of the chain–alkane substituent groups appropriately facilitated an improvement in activity, such as for compound **9b** (R = propylamine, 22.0 µM) > **9a** (R = butylamine, 20.32 µM). The IC_50_ values of the halogen-substituted compounds **9d**–**9g** were lower than that of compound **9c** when the substituted group was the benzene ring. Above all, compound **5i** exhibited the best anti-tumor activity against A549 with an IC_50_ value of 6.15 µM, indicating that the 3-F substituent facilitated the enhancement of anti-tumor activity. Hence, **5i** was selected as the candidate compound for further pharmacological study to explore its anticancer mechanisms.

### 2.4. Interaction of EGFR with Compound ***5i***

The binding mode between the EGFR (Epidermal Growth Factor Receptor) (PDB: 1M17) and compound **5i** is shown in Figure 4A,B. The simulation results indicated that **5i** exhibited its binding ability by penetrating into the binding bag of the EGFR. In the binding bag, the oxygen atom in the naphthoquinone ring could form hydrogen bonds with the nitrogen atom of Asp (aspartic acid) 831 in the EGFR and the hydrogen bond of the oxygen atom of Thr (threonine) 830 in the EGFR, respectively (Figure 4C,D). In addition, the benzene ring on the naphthoquinone of compound **5i** was able to interact with the carbon atom of Lys (lysine) 721 to form pi-H (Figure 4C,D). These interactions between EGFR and **5i** mainly changed the binding energy and had impacts on the EGFR-associated pathway in vivo.

### 2.5. Compound ***5i*** Induced ROS Accumulation in A549 Cell

Naphthoquinone derivatives were reported that induced the intracellular ROS (reactive oxygen species) accumulation to affect the therapeutic effect [53]. To investigate the potential relationship between compound **5i** and ROS production, we treated A549 cells with **5i** at different concentrations for 6 h and 12 h, and observed the changes of ROS by flow cytometry. At the earlier stage (6 h), poor ROS accumulation was observed both in **5i**-treated and control groups. This result might indicate that compound **5i** had not yet had enough time to induce large numbers of ROS production. When the incubation time was 12 h, the ROS level in A549 cells significantly increased compared to the negative controls, which was nearly a three-fold increase (Figure 5A,B). As shown by the high levels of ROS, we speculated that compound **5i** inhibited the proliferation of cancer cells by inducing ROS accumulation.

### 2.6. Effect of ***5i*** on EGFR Signal Pathway-Related Proteins

To further investigate the inhibitory mechanism of compound **5i** on A549 cells, firstly, we studied the impact of **5i** on the EGFR-associated signal pathway through detecting the expression of EGFR, AKT, and their phosphorylated proteins at 10 and 20 μM, respectively, by the Western blot assay. As shown in Figure 6A,B, we found that the proteins expression was obviously different between the **5i** treatment group and the blank control group. Compound **5i** significantly up-regulated the expressions of p-Akt, which were translated by tumor suppress genes, and the levels of p-EGFR and Akt remained unchanged regardless of treatment with **5i** or gefitinib at 10 μM. Interestingly, EGFR protein expression was down-regulated remarkably after treatment with **5i** (10 μM), which lower three times lower than the control group. However, little fluctuation of the EGFR was observed after treating with 10 μM gefitinib, which indicates that compound **5i** playa a distinctive role in the regulation of the EGFR pathway to affect A549 cells. Hence, combined with the above results, we preliminarily suspect that compound **5i** will induce autophagy by promoting EGFR recycling and signal transduction in A549 cells to activate the EGFR signaling pathway. 

### 2.7. The Effect of Compound ***5i*** on the Expressions of the Autophagy-Related Protein LC3

Autophagy has been found to play a key role in physiological and pathological processes in vivo, and the appropriate modification of autophagy could accelerate the apoptosis process of tumor cells and enhance the efficacy of chemotherapy [54,55,56]. It has been known that LC3 proteins are associated with autophagy in vivo [57,58]. To verify whether compound **5i** could affect the autophagy process in A549 cells, this work studied the expression of the autophagy-related protein LC3 in Western blotting experiments. After treatment with **5i** for 48 h, we found that LC3-I down-regulated significantly, and on the contrary, LC3-II was up-regulated twice at 10 and 20 μM compared control group (Figure 7A,B). However, the positive drug gefitinib seemed to have little influence on the expression of LC3-II in the A549 cells. Remarkable changes between LC3-I and LC3-II after treatment with **5i** indicated that **5i** might play a vital role in inducing the death of A549 cells through autophagy to achieve a good anti-tumor effect. 

## 3. Discussion

Autophagy is an important metabolic regulation mechanism, which can remove dysfunctional organelles and extra macromolecules to protect cells against damage from various stress reactions [59,60]. An increasing number of evidence has shown that autophagy plays a critical role in diverse human physiological and pathological processes, including tumors and infections [61]. A great variety of investigations demonstrated that tumor cells with defects in the autophagy facilitated a malignant transformation [62], indicating that autophagy might be a potential anti-tumor target, which was worthy of further study and exploration to develop promising drug molecules. In recent years, small chemical molecules became an important tool for treating cancers and exploring the basic biological mechanisms and processes of malignant tumors [63,64,65,66,67,68,69].

Overall, naphthoquinone derivatives have shown anti-cancer effects against various cancers through a variety of mechanisms. Most reported strategies for the development of anti-tumor drugs were focused on DNA damage through ROS production, the inhibition of topoisomerase II, the inhibition of tumor suppressor p53, and the induction of apoptosis through ERS. In this study, we synthesized a series of new 1,4-naphthoquinone derivatives containing two amino substituents, and screened a promising compound **5i** by in vitro biological activity testing, followed by an investigation of its anti-cancer mechanisms. We found that the inhibitory activity of **5i** on the growth of A549 cells might have resulted from autophagy, based on the regulation of LC3 proteins expression that was associated with autophagy. Our research results indicated that autophagy could be a potential anti-tumor target of the two amino-substituted 1,4-naphthoquinone derivatives. In this work, an interesting experimental phenomenon that was different from the widely reported mechanism of naphthoquinone derivatives was found and analyzed. 

It is worth mentioning that doxorubicin (ADR), a class of anthracycline antibiotics, is a highly effective, broad-spectrum, anti-tumor antibiotic that is widely used in the clinical treatment of breast cancer, lung cancer, lymphoma, and other malignant tumors. However, ADR is limited to cardiomyopathy treatment based on its toxicity, which results in congestive heart failure and death. Various speculations on the ARD induction of cardiotoxicity were proposed, and the oxygen-free radical theory was accepted. ARD is supposed to be involved in the key cellular REDOX process that produces large numbers of ROS, which leads to the modification and degradation of nucleic acids and proteins in cells. To avoid these side effects, ARD is usually enclosed in drug carrier, taking the liposomes as an example, which deliveries ARD to the desired organs and alters the pharmacological distribution of the drug to avoid damage of the heart by an excess of ARD. In addition, it has also been reported that ARD-liposome enabled the preferential accumulation of drugs in target tumors, so that the performance of ARD was enhanced dramatically [70]. 

Similarly, quinone derivatives are widely used as anti-oxidants or mediators ascribed to the formation of ROS with superoxide anion radicals, hydrogen peroxide, hydroxyl radicals, and singlet oxygen [71]. Based on the redox characteristic of naphthoquinone derivatives, in this study, we studied the generation of ROS in A549 cells that were induced by compound **5i** via flow cytometry, and we found that the ROS levels increased significantly after 12 h of co-incubating with compound **5i**. In future studies, compound **5i** should be enclosed in liposomes or modified by specific peptides to avoid the potential side effects and targets to the diseased organs.

## 4. Materials and Methods

### 4.1. Chemistry and Instruments

All the starting materials, reagents, and solvents for synthesis of target compounds are commercially available unless otherwise specified. All the reactions were monitored by thin layer chromatography (TLC) on a silica gel plate (GF254) and observed under ultraviolet light (254 and 365 nm). All solvents and reagents were reagent-grade without further purification. NMR data were obtained by Bruker Ascend 400 and 500 MHz (Bruker Optics, Switzerland). The following abbreviations are used to describe peak-splitting modes: s (singlet), d (two-state), t (tristate), m (polymorphic), and dd (two-state of two-state). The coupling constant is expressed in Hertz units (Hz). For the single-crystal data of compound 5a, the X-ray diffraction data of single crystal were recorded by Bruker D8 QUEST diffractometer and analyzed by Olex-2 software. High resolution mass spectrometry (HRMS) spectra were conducted using a Thermo Scientific Q Exactive (Thermo Scientific, St. Louis, MO, USA). The melting points (m. p.) for all title 1,4-naphthoquinone derivatives were detected using a (X-4D) digital micro melting point apparatus.

### 4.2. Pharmacology

#### 4.2.1. Cell Culture and Treatment

Non-small human lung cancer (A549), chronic myeloid leukemia (K562), prostate (PC3), and liver (HepG2) cells were purchased from the Cell Bank of the Chinese Academy of Sciences (Kunming, China). A549, K562, and PC3 cells were cultured in RPMI-1640 medium and HepG2 were cultured in MEM medium. All mediums were supplemented with 10% fetal bovine serum (FBS), 100 μg/mL penicillin, and 100 μg/mL streptomycin. All cells were cultivated in 5% CO_2_ incubator at 37 °C.

#### 4.2.2. MTT Assay

According to the MTT method, the inhibitory activities of 1,4-naphthoquinone derivatives on PC-3 cells, HepG2 cells, K562 cells, and A549 cells were detected after treatment in different concentration ranges (1–30 µM) for 48 h. The specific procedures are as follows: Then, cancer cells at logarithmic growth stage were digested with 0.25% trypsin and re-suspended in DMEM and RPMI culture medium containing 10% FBS. Cancer cells were inoculated into 96-well plates (cell concentration: 2 × 10^4^ cells/mL) at a volume of 100 µL per well, with blank control group on the left. Cell-free RPMI medium with serum was added and placed in an incubator at 37 °C with 5% concentration of carbon dioxide and saturated humidity. Cells were required to be in a state of adherent growth for 24 h. After 24 h, the medium was removed and added to the corresponding hole of the 96-well plate at 200 µL per well (the final concentration of DMSO was less than 0.1%), containing different drugs to be tested. The blank control group was added with 200 µL per well of complete medium. Under the same experimental conditions, 48 h later, the supernatant was poured away, and each well in the 96-well plate was incubated at room temperature (25 °C) for 4 h. DMSO was added to the 96-well plate with 150 µL per well. At room temperature, the cells were shaken by cell shaker for 5 min. The OD value (absorption wavelength 490 nm) was determined with enzyme label, and then the inhibition rate of compound on cells was calculated. Three parallel experiments should be conducted for each sample, with the concentrations of six holes parallel each time, and the calculated average is the final result of the experiment. The relative cell viability was calculated according to the following equation:$$\mathrm{Cell}\;\mathrm{viability}\left(\%\right)=\frac{\mathrm{experimental}\;\mathrm{OD}\;\mathrm{value}}{\mathrm{control}\;\mathrm{OD}\;\mathrm{value}}\times 100$$
$$\mathrm{Cell}\;\mathrm{inhibition}\left(\%\right)=\frac{\mathrm{control}\;\mathrm{OD}\;\mathrm{value}-\mathrm{experimental}\;\mathrm{OD}\;\mathrm{value}}{\mathrm{control}\;\mathrm{OD}\;\mathrm{value}}\times 100$$

#### 4.2.3. Western Blotting Analysis

The 10^4^/well cells in the 6-well plates were cultured for about 24 h and incubated with the compound for 48 h. After cleaning with PBS, cells were added into the mixture of cell lysate RIPA (Beyotime, Shanghai, China), protease inhibitor (Beyotime, Shanghai), and phosphatase inhibitor (Beyotime, Shanghai) 50 μL/well. The cells in the well plates were scraped off with a cell scraper and collected in a small centrifuge tube. The cells were cleaved fully at 4 °C for 1 h. After centrifuging at 4 °C for 15 min (12,000 rpm), supernatant was taken to obtain total cell protein. BCA protein concentration assay kit (Biyuntian, Shanghai) was used to quantify the protein. After incubation at 37 °C for half an hour, the absorbance was measured at 562 nm (in the enzyme marker). Standard diagram (c-OD diagram) was made from 8 protein standards with known concentrations to obtain the required protein sample concentration, which was prepared and marked. Then, they were boiled at 100 °C in a water bath for 10 min for denaturation, and stored at 4 °C for later use. The protein was isolated by SDS-PAGE and then transferred to PVDF membrane. After the membrane transfer was complete, the PVDF membrane was cut according to marker and its molecular weight, immersed in the sealing liquid (5% BSA), and closed at room temperature for 1h with uniform shock (shaking bed 40 rpm). After the closure was complete, the corresponding antibody was added and incubated for 2 h. The antibody could be incubated at 4 °C overnight. Then, the second antibody was added and incubated for 1h. After the incubation, TBST buffer was added 3 times and washed for 5 min each time. According to the size and quantity of the membrane, an appropriate amount of chemiluminescent solution was prepared and dropped on the membrane. The color developed in the gel imager (VilberLourmat), and the data were analyzed by ImageJ software (Version 1.53t 24 August 2022).

The antibodies against EGFR (AF6043, 1:1000) and β-actin (AF7010,1:10,000) were purchased from Affinit. The antibody against LC3 (14600-1-AP, 1:2500) and GAPDH (10494-1-AP, 1:20,000) was purchased from Proteintech. The antibodies against P-EGFR (3777S, 1:1000) and P-AKT (4058T, 1:1000) were purchased from Cell Signaling.

#### 4.2.4. Molecular Docking Study

EGFR (PDBID code: 1M17) eutectic structure model was used for docking. MOE (2019) was used to predict the binding pattern of ligands to the corresponding EGFR (PDBID code: 1M17). Protein structures can be downloaded from the website https://www.rcsb.org (accessed on 1 January 2023) and processed using the protein preparation wizard. The selected compounds were ligands and they were treated with QuickPrep. The whole kinase was used as the receptor and the processed ligand was selected for docking. These image files were generated using MOE (2019).

#### 4.2.5. Statistical Analysis

Statistical processing of results was performed using GraphPad Prism 8.0 software (GraphPad Inc., San Diego, CA, USA). All data were expressed as mean ± SD and all the data provided were verified by at least three independent experiments. Differences among groups were considered significant at *p* ≤ 0.05.

## 5. Conclusions

In conclusion, this study reported the synthesis and anti-proliferative activities of two amino-substituted 1,4-naphthoquinone derivatives. Most compounds showed good biological activities on the tested tumor cells, especially compound **5i** (IC_50_ = 6.15 ± 0.19 µM), which exhibited significant anti-proliferation activity against the A549 cells. Based on the preliminary anti-tumor mechanism experimental results, **5i** is a promising leading compound that can induce autophagy by promoting EGFR recycling and signal transduction, and therefore, it can provide a new research path for the development of effective autophagy inhibitors.

## Figures and Tables

**Figure 1 molecules-28-03289-f001:**
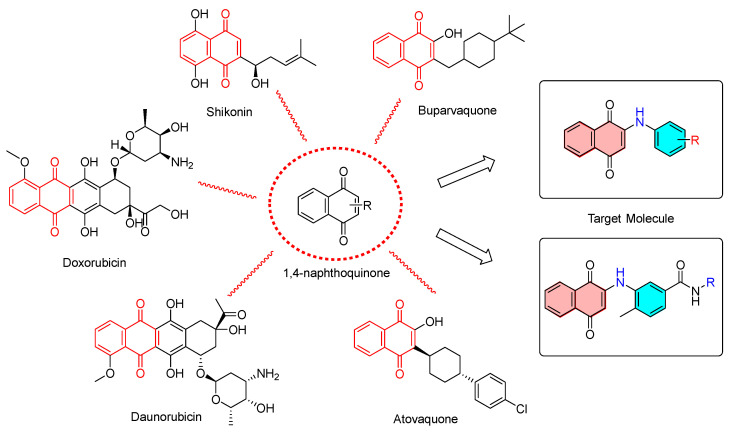
The chemical structures of representative 1,4-naphthoquinone anti-cancer drugs.

**Figure 2 molecules-28-03289-f002:**
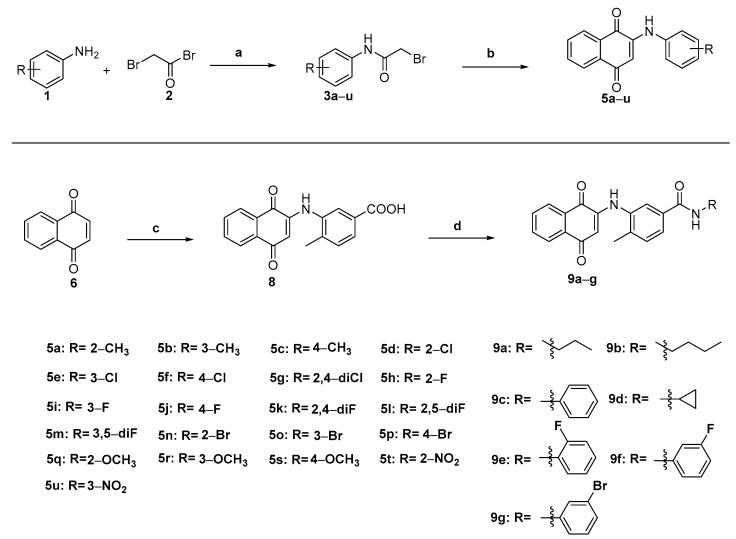
Synthetic route of target molecules. Reagents and conditions. (a) K_2_CO_3_ and DCM (Dichloromethane) at room temperature. (b) 2-Hydroxy-1,4-naphthoquinone, Cs_2_CO_3_, and DMF at room temperature. (c) 3-amino-p-toluidic acid, Cu(OAc)_2_, and AcOH at 60–70 °C. (d) Different substitution of fatty amines and aromatic amines, HATU, Dipea, and DCM at room temperature.

**Figure 3 molecules-28-03289-f003:**
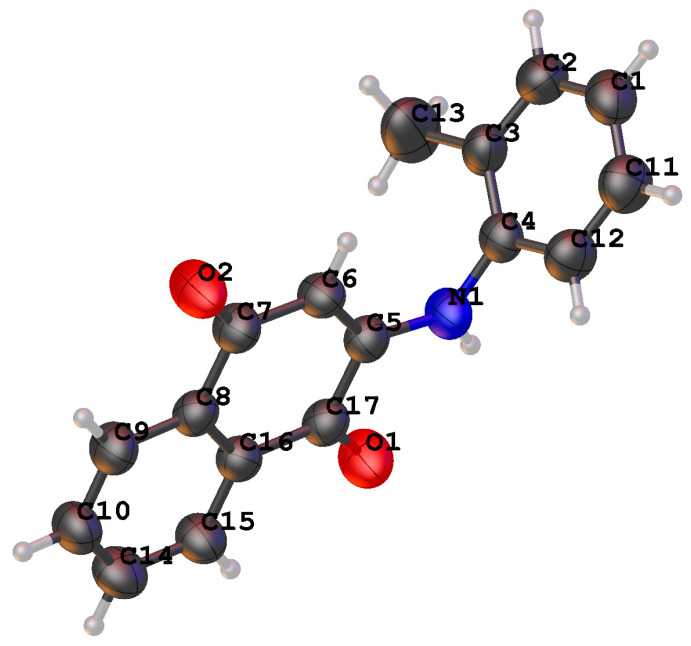
The crystal structure of compound **5a**.

**Figure 4 molecules-28-03289-f004:**
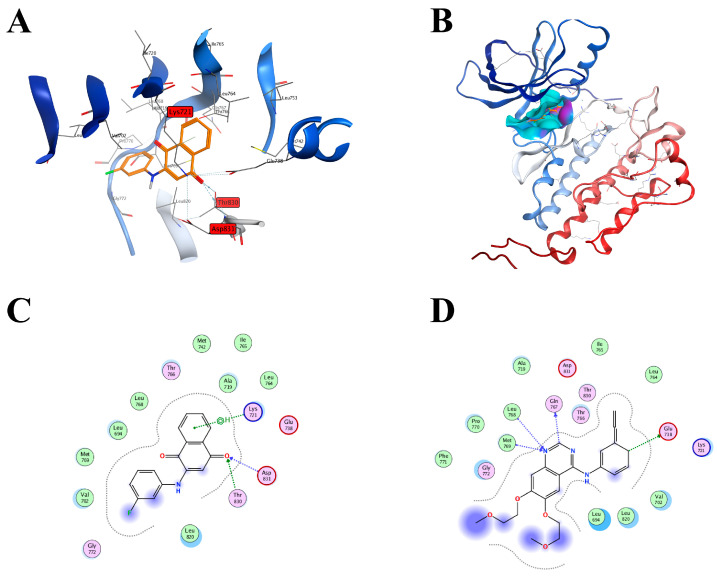
The binding mode of EGFR and compound **5i** or Erlotinib. (**A**) The three-dimensional binding mode of EGFR and compound **5i**: compound **5i** is orange, residues around the binding bag are displayed in gray, and hydrogen bonds are indicated by dotted lines in light blue. (**B**) Surface binding mode of EGFR to compound **5i**. (**C**) Two-dimensional binding mode of EGFR and compound **5i**. (**D**) Two-dimensional binding mode of EGFR and Erlotinib.

**Figure 5 molecules-28-03289-f005:**
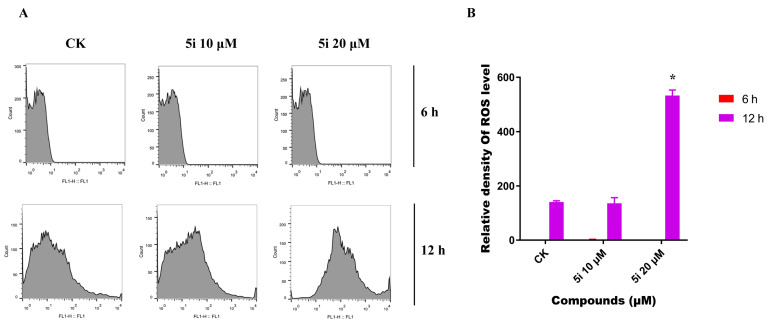
Compound **5i** induces ROS accumulation in A549 lung cancer cells. (**A**) A549 cells were treated with 10 μM and 20 μM **5i** at different time points (6 or 12 h) and stained with DCFH-DA (10 μM). Intracellular ROS levels were determined by flow cytometry. (**B**) A549 cells treated with 10 μM and 20 μM compound **5i** at different times (6 or 12 h) were quantitatively analyzed. The values are expressed as mean SD from three individual experiments. * Statistically significant (*p* < 0.05).

**Figure 6 molecules-28-03289-f006:**
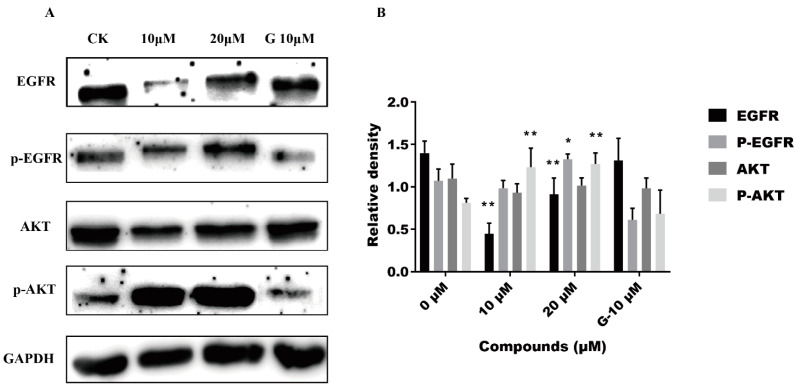
The effect of **5i** on EGFR protein kinase signal pathway. (**A**) A549 cells were incubated with **5i** of different concentrations for 48 h. Gefitinib with a concentration of 10 µM was used as positive control (G-10 µM). Expressions of EGFR, P-EGFR, Akt, and P-Akt were detected by Western blot. (**B**) Quantify the relative protein expression levels of EGFR, P-EGFR, Akt, and P-Akt. GAPDH is used for load control. The values are expressed as mean SD from three individual experiments. * Statistically significant (*p* < 0.05). ** Statistically extremely significant (*p* < 0.01).

**Figure 7 molecules-28-03289-f007:**
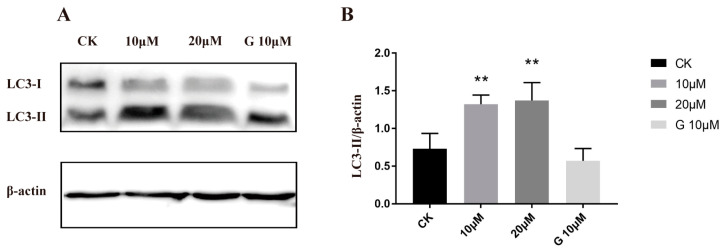
The effects of **5i** on the autophagy-related protein LC3. (**A**) Western blot analysis of LC3-I and LC3-II of A549 cells treated with **5i** (0, 10, and 20 μM) for 48 h. Gefitinib with a concentration of 10 µM was used as positive control (G-10 µM). (**B**) The relative protein expression levels of LC3-I and LC3-II were quantified by normalization to β-actin. β-actin was used as a loading control. The values are expressed as mean SD from three individual experiments. ** Statistically extremely significant (*p* < 0.01).

**Table 1 molecules-28-03289-t001:** The anti-proliferative activity of 1,4-naphthoquinone derivatives on cell carcinoma lines after 48 h treatment.

Compounds	IC_50_ ± SD ^1^ (μM)				
	R	A549	K562	HepG2	PC-3
**5a**	2-CH_3_	8.59 ± 0.50	-	-	-
**5b**	3-CH_3_	8.14 ± 0.34	-	-	-
**5c**	4-CH_3_	> 50	-	-	-
**5d**	2-Cl	20.32 ± 6.43	-	-	-
**5e**	3-Cl	23.45 ± 7.54	-	-	14.49 ± 1.38
**5f**	4-Cl	> 50	-	-	-
**5g**	2,4-diCl	14.11 ± 1.68	-	-	-
**5h**	2-F	23.59 ± 7.21	-	-	34.61 ± 6.45
**5i**	3-F	6.15 ± 0.19	-	-	-
**5j**	4-F	19.03 ± 3.86	17.79 ± 3.21	26.42 ± 3.05	18.4 ± 3.46
**5k**	2,4-diF	17.23 ± 4.89	-	20.15 ± 2.49	-
**5l**	2,5-diF	7.99 ± 1.27	-	23.59 ± 5.27	23.78 ± 5.94
**5m**	3,5-diF	6.83 ± 0.72	25.61 ± 9.62	-	-
**5n**	2-Br	26.58 ± 8.98	-	23.64 ± 5.08	9.71 ± 1.87
**5o**	3-Br	21.23 ± 3.55	26.60 ± 6.18	-	-
**5p**	4-Br	8.34 ± 3.13	19.53 ± 7.86	-	10.31 ± 0.82
**5q**	2-OCH_3_	10.44 ± 0.52	-	-	-
**5r**	3-OCH_3_	27.76 ± 3.56	-	-	-
**5s**	4-OCH_3_	16.77 ± 2.81	29.62 ± 8.90	-	19.34 ± 4.74
**5t**	2-NO_2_	8.45 ± 2.15	-	-	-
**5u**	3-NO_2_	15.64 ± 1.87	-	-	-
**9a**	propylamine	20.32 ± 6.70	-	-	-
**9b**	butylamine	22.00 ± 3.06	21.42 ± 4.02	17.90 ± 1.64	-
**9c**	phenyl	35.55 ± 7.98	-	-	-
**9d**	cyclopropyl	16.54 ± 3.83	-	-	-
**9e**	2-fluorophenyl	14.32 ± 4.18	-	-	-
**9f**	3-fluorophenyl	23.03 ± 4.81	24.07 ± 2.67	-	-
**9g**	3-bromophenyl	21.31 ± 6.59	-	22.36 ± 1.45	15.49 ± 1.55
	G ^2^	22.11 ± 2.32	15.64 ± 2.59	30.67 ± 11.30	12.02 ± 5.40
	5-Fu ^3^	1.34 ± 0.32	4.01 ± 0.21	12.05 ± 5.27	8.95 ± 0.87

^1^ IC_50_ values are the concentrations that cause 50% inhibition of cancer cell growth. Data represent the mean values ± standard deviation of three independent experiments performed in triplicates. ^2^ Gefitinib (G) is the reference standard. ^3^ 5-Fluorouracil (5-Fu) is the reference standard.

## Data Availability

Not applicable.

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
