# Peer review of "Novel 2-Amino-1,4-Naphthoquinone Derivatives Induce A549 Cell Death through Autophagy"

_molecules, 2023, doi:10.3390/molecules28083289_

Round 1

Reviewer 1 Report

Tan, et al. synthesized a series of 2-amino substituted 1,4-naphthoquinone derivatives, and found that compound 5i show significant cytotoxicity on A549 cell line with the IC50 of 6.15 μM. Based on the anti-tumor mechanism study, they demonstrated that the representative compound 5i could induce autophagy by promoting EGFR recycling and signal transduction in A549 cells. In addition, the confirmation of single crystal structure of compound 5a helped the further understand other target compounds and analyze their structure-activity relationships. Their work provided a new research way for development of effective autophagy drugs that induce A549 cell death. The results were clearly presented, and experimental conditions were described in detail. I would recommend considering acceptance of the manuscript subject to a couple of minor revisions.

1. Please provide a higher resolution single crystal image of compound 5a, as Figure 3 in the manuscript is not clear enough.

2. The description of cesium carbonate reaction in Figure 2 comment (b) is inappropriate. Please revise it.

3.      Some journal names in the reference parts are not abbreviated, and the format of references should be more accurate, standardized and uniform.

4.      A more in depth discussion of structure-activity relationships should be presented in the manuscript.

5.      In the synthesis route of 5a-5u, please explain why this paper obtained target compounds 5a-5u, rather than the products that contained amide group.

Reviewer 2 Report

Main objections:

1. The main characteristic of quinones in chemical and biological systems is their redox activity, i.e. the ability to be reduce and oxidize. As a result, they actively generate reactive oxygen species (ROS) [1]. These ROS are the basis of both therapeutic and side effects of these quinones. The redox activity of the investigated quinones was not studied in any way (and not even mentioned!) in this work?!.
2. To avoid side effects of such compounds, their conjugates with biological addresses have long been produced [1] or, like doxorubicin, they are enclosed in liposomes [2}, i.e. there is a targeted delivery to the desired (diseased) organs. These approaches are not available in this work.

I believe that without these data the modern work on medical application of  quinones cannot be published.   Additional consideration: doxorubicin is not a naphthoquinone!   I  1 .Shai RahimipourGeorg GescheidtItzhak BilkisMati Fridkin & Lev Weiner Towards the Efficiency of Pharmacologically Active Quinoid Compounds: Electron Transfer and Formation of Reactive Oxygen Species.Applied Magnetic Resonance  37 629–648 (2010) 2.A.Abraham, Dawn N. Waterhouse, Lawrence D. Mayer, Pieter R. Cullis, Thomas D. Madden, Marcel B. Bally The Liposomal Formulation of Doxorubicin Methods in Enzymology, 391,71-97 (2005)

Reviewer 3 Report

The set of 1,4-naphthoquinone derivatives 5a-5u and 9a-9g, were synthesized and characterized using 1H NMR, 13C NMR, and HRMS techniques. The preliminary antiproliferative activity results showed that compound 5i exhibited significantly anti-proliferation activity against A549 cells. Compound 5i can be considered as a promising leading compound that induces autophagy by promoting EGFR recycling and signal transduction.

This work should be improved according to the following comments:

1) L.82: Figure 2. Yields should be added for all products. L. 83: it should be “2-Hydroxy-1,4-naphthoquinone” instead of “2-Hydroxy-1,4-naphoquinone”.

2) L.119: The sentence should be started as “The compound 5i…”

3) Clarify the meaning of the “-“ sign in Table 3.

4) Add the p values for Figure 5B.

5) L. 218: Check the sentence, it should be about 1,4-naphthoquinone derivatives.

6) In the section 4.1 you should add the description of X-Ray diffractometer.

7) What does the “(R)” stand for in compounds name (Table 1 and 2)?

8) The reaction between 3a-3u and 2-hydroxy-1,4-naphthoquinone in the presence of Cs2CO3 is unclear. Could you provide possible mechanism of reaction and references to similar transformation?

9) You must clearly specify the compounds that have been previously reported in The Journal of Organic Chemistry, 2011, 76(13), 52645273. Also refer to this work.

10) How did you determine the purity of the target compounds?

11) In supplementary material:

Some NMR spectra are very poor quality, for example, Fig S23, Fig S25, Fig S46, Fig S50, etc. Could you provide spectra with better quality? Figure S15 and S17 are the same, also S16 and S18.

Correct the scale on the f1 and f2 axes to fit spectrum size for all NMR spectra.

Reviewer 4 Report

The article "Novel derivatives of 2-amino-1,4-naphthoquinone induce A549 cell death by autophagy" describes the synthesis and in vitro biological evaluation of a series of novel anticancer compounds. The work was well planned and described, and the results obtained are interesting.

However, here are some issues that need to be addressed before an article is accepted:

1.      A549 is non-small human lung cancer not “small non-human lung cancer” (lines 60 I 233)

2.      English needs checking (e.g. lines 107-108 “Pen-tafluor ouracil 5-fluorouracil”, line 234 “liver” instead of “live”).

3.      Paragraphs from lines 87-95 are a repetition of the methodology section.

4.      Table 3 - Have IC50 values been tested for compounds with dashes in the table? Were the IC50 values of compounds 5i, 5m, 5o tested against the PC3 line, or 9b and 9e against the K562 line? Preliminary values in Tables 1 and 2 are promising for these compounds.

5.      Moving Tables 1 and 2 to the supplement may be considered, instead a column with substituents of the tested compounds may be added to Table 3. The headings of tables 1-3 can be extended with information about the culture time (48 h).

6.      The caption of figure 1 you can be extended with "anticancer drugs" instead of "drugs" (line 66).

7.      It is worth adding extending the information about G-10uM in the description of figures 5 and 6.

8.      The supplementary information requires corrections of text formatting and punctuation (missing spaces, e.g. 1HNMR, missing superscripts 1H NMR, 13C NMR, H+, missing subscripts in compound formulas).

9.      Please explain DCM abbreviation.

1.   There is Chinese description under Figure S44.  

1.   How can the presence of additional signals in the HRMS spectra for compounds 5m, 9a, 9c and 9e be explained?

Round 2

Reviewer 2 Report

The corrected version of the work looks better. However, it again lacks a discussion of the possible specific delivery of naphthoquinone to the "diseased" organ: conjugation with specific peptides (proteins), the use of nanoparticles, etc. These and other important issues are discussed in the two links we mentioned in the first review. These references are missing from the work and should be included in the manuscript.
